# Antihypertensive Activity of the Alkaloid Aspidocarpine in Normotensive Wistar Rats

**DOI:** 10.3390/molecules27206895

**Published:** 2022-10-14

**Authors:** Noemi Oliveira Monteiro, Theresa de Moura Monteiro, Thalya Soares R. Nogueira, Jacqueline Rodrigues Cesar, Lara Pessanha S. Nascimento, Karoline Azerêdo Campelo, Graziela Rangel Silveira, Fernanda Antunes, Daniela Barros de Oliveira, Almir Ribeiro de Carvalho Junior, Raimundo Braz-Filho, Ivo José Curcino Vieira

**Affiliations:** 1Laboratório de Tecnologia de Alimentos, CCTA, Universidade Estadual do Norte Fluminense Darcy Ribeiro, Campos dos Goytacazes, Rio de Janeiro 28013-602, Brazil; 2Laboratório de Clínica e Cirurgia Animal, CCTA, Universidade Estadual do Norte Fluminense Darcy Ribeiro, Campos dos Goytacazes 28013-602, Brazil; 3Laboratório de Ciências Químicas, CCT, Universidade Estadual do Norte Fluminense Darcy Ribeiro, Campos dos Goytacazes, Rio de Janeiro 28013-602, Brazil; 4Instituto Federal de Educação, Ciência e Tecnologia da Bahia, Vitória da Conquista, Bahia 45078-300, Brazil; 5Departamento de Química Orgânica, Instituto de Química, Universidade Federal Rural do Rio de Janeiro, Seropédica, Rio de Janeiro 20000-000, Brazil

**Keywords:** *Aspidosperma desmanthum*, alkaloid, Aspidocarpine, blood pressure, rotarod

## Abstract

The alkaloid Aspidocarpine was isolated from the bark of *Aspidosperma desmanthum*. Its structure was elucidated by the spectral data of ^1^H and ^13^C-NMR (1D and 2D) and high-resolution mass spectrometry (HRESIMS). The antihypertensive activity was investigated by intravenous infusion in Wistar rats. This alkaloid significantly reduced (*p* < 0.05) the systolic, median, and diastolic blood pressures of rodents, without causing motor incoordination and imbalance in the rotarod test. The results indicate that the alkaloid Aspidocarpine exerts its antihypertensive activity without causing sedation or the impairment of motor functions.

## 1. Introduction

Hypertension is a major public health problem and the leading cause of cardiovascular mortality worldwide, mainly affecting people living in low- and middle-income countries [1]. Although this disease can be controlled through pharmacological interventions, the drugs commonly used can present several side effects and have a high cost. [2]. In this sense, medicinal plants represent a good alternative, as many have demonstrated hypotensive capacity, cause few side effects, and have a low cost. [3,4].

The genus *Aspidosperma* (Apocynaceae) is composed of approximately 50 species, all restricted to the tropical and subtropical regions of America, where they are popularly known as “peroba” and “guatambu” [5,6,7]. Many of this species are used in folk medicine for the treatment of malaria [8], fever [9], inflammation, diabetes, high blood pressure [10], and cardiovascular diseases [11]. In addition, studies carried out with extracts from several species of *Aspidosperma* showed important antihypertensive effects [12,13,14,15].

The biological activities presented by plant species have been associated with the occurrence of secondary metabolites [16]. In *Aspidosperma*, the most abundant secondary metabolites are indole alkaloids, to which several biological activities are attributed [17], such as antiplasmodial [18], antiprotozoal [19], cytotoxic [20], anti-inflammatory [21], and antihypertensive [22].

Considering the biological importance of alkaloids, the antihypertensive activity of the alkaloid Aspidocarpine, isolated from the bark of *A. desmanthum*, was determined. This work presents the first report on the antihypertensive activity of the alkaloid Aspidocarpine.

## 2. Results and Discussion

### 2.1. Identification and Structure of the Compound

The chromatographic fractionation of the dichloromethane partition, obtained through the methanolic extract of the bark of *Aspidosperma desmathum*, led to the isolation of the alkaloid Aspidocarpine (Figure 1). The chemical profile was then analyzed by mass spectrometry, and NMR analysis (Appendix A), and by comparing its spectral data from ^1^H-, ^13^C-NMR with values from the literature.

Aspidocarpine was obtained as a yellow crystal. ^1^H-NMR (CDCl_3_, 500 MHz): *δ*_H_ 10.91 (s, OH-12), 6.60 (d, *J* = 8.1, H-10), 6.52 (d, *J* = 8.1, H-9), 4.03 (dd, *J* = 11.3, 6.2, H-2), 3.77 (s, OMe-11), 3.06 (td, *J* = 9.0, 2.1, H-5a), 2.97 (br d, *J* = 10.9, H-3a), 2.23 (s, 3H-2′), 2.20 (s, H-21), 2.20 (m, H-14a), 2.15 (m, H-5b), 1.98 (m, H-6a), 1.98 (m, H-17a), 1.92 (m, H-14b), 1.90 (m, H-3b), 1.75 (m, H-16a), 1.60 (m, H-15a), 1.45 (m, H-6b), 1.40 (m, H-16b), 1.25 (m, H-19a), 1.02 (m, H-15b), 1.02 (m, H-17b), 0.80 (m, H-19b) and 0.64 (t, *J* = 7.4, 3H-18). ^13^C-NMR (CDCl_3_, 125 MHz): *δ*_C_ 169.2 (C-1′), 149.2 (C-11), 137.3 (C-12), 132.9 (C-8), 127.4 (C-13), 112.3 (CH-9), 109.9 (CH-10), 70.6 (CH-21), 69.3 (CH-2), 56.4 (OMe-11), 53.5 (CH_2_-3), 52.2 (CH_2_-5), 51.9 (C-7), 39.1 (CH_2_-6), 35.7 (C-20), 33.8 (CH_2_-15), 30.0 (CH_2_-19), 24.9 (CH_2_-16), 22.7 (CH_2_-17), 22.6 (CH_3_-2′), 21.2 (CH_2_-14), 7.0 (CH_3_-18). The molecular formula was confirmed to be C_22_H_30_N_2_O_3_ by its HRESIMS (*m/z* = 371.2324 [M + H]^+^, C_22_H_31_N_2_O_3_). The data are in accordance with those previously published [23,24,25,26,27].

### 2.2. In Vivo Blood Pressure Assessment

Indole alkaloids are the main secondary metabolites produced in *Aspidosperma* and are responsible for most of the biological activities reported in the genus, including antihypertensive activity. An intravenous infusion of the alkaloid Aspidocarpine (1 and 3 mg/kg) was administered in rats to investigate whether this compound can cause changes in blood pressure. After the administration of a 1 mg/kg infusion of Aspidocarpine, systolic and diastolic pressures were significantly reduced in relation to the negative controls and a positive control (DMSO), and after the administration of a 3 mg/kg infusion, all the parameters analyzed were significantly reduced in relation to the negative controls and a positive control (DMSO) in Wistar rats at a 5% probability (Figure 2).

The Tubotaiwine alkaloid, commonly isolated from *Aspidosperma* [28], significantly reduced the cadmium-induced increase in systolic and diastolic blood pressure in rats at doses of 2.5, 5, and 10 mg/kg [22]. In the present study, Aspidocarpine showed a similar antihypertensive response, but in lower concentrations. In addition, extracts of *A. subincanum* [15], *A. pyrifolium* [14], *A. fendleri* [13], and *A. macrocarpum* [12] showed significant hypotensive effects, consistent with the results presented by Aspidocarpine. These reports confirm that the genus *Aspidosperma* is an important source of alkaloids with cardiovascular activity.

### 2.3. Rotarod Test

To assess the possible effects of Aspidocarpine on motor performance, the rotarod test was used. An intraperitoneal administration of 1 mg/kg and 3 mg/kg Aspidocarpine did not affect the time the mice remained on the rotating bar. In contrast, diazepam (positive control) significantly reduced the locomotor activity and performance on the rotarod (Table 1).

The treatment with Aspidocarpine did not produce motor incoordination, hypolocomotion, immobility, clinical signs of sedation, or a stimulatory effect in mice, and did not cause lethality at the doses studied. This result is in agreement with previous studies that reported that the ethanolic extract of *A. nitidum* did not reduce the time spent by the animals on the rotating bar, indicating that the antinociceptive action presented was not related to neurological or motor alterations [29]. In addition, the alkaloid-rich extract of *A. ulei* induced an increase in motor locomotion, but did not cause motor impairment in the rotarod test in contrast to diazepam, indicating a central nervous system stimulatory effect [30]. Furthermore, the extract of *A. pyrifolium* seeds demonstrated a neuroprotective effect by reducing motor incoordination in a rat model of Parkinson’s disease [31].

The performance of Aspidocarpine in the rotarod test was similar to that of other alkaloids that showed neuroprotective activity associated with anti-inflammatory activity [32,33]. In addition, the ability of some alkaloids to inhibit the release of inflammatory factors was also related to cardioprotective activities [2,34]. Thus, future studies that evaluate the anti-inflammatory activity of Aspidocarpine are necessary to determine the possible mechanism of action of this antihypertensive activity.

The present work is the first to report the influence of an alkaloid isolated from the genus *Aspidosperma* on motor coordination.

## 3. Materials and Methods

### 3.1. General Experimental Procedures

^1^H- (500 MHz) and ^13^C- (125 MHz) NMR data were obtained on a Bruker Advance II 9.4 T instrument (Centro de Ciências e Tecnologia, UENF) using deuterated chloroform as a solvent. HR-ESI-MS mass spectra was obtained on a micrOTOF-Q II Bruker Daltonics mass spectrometer (Billerica, MA, USA) using the positive ion mode of analysis. Chromatographic purifications were performed using silica gel 60 (0.063–0.200 mm, MERCK, Darmstadt, Germany). Aluminum-backed Sorbent silica gel plates, w/UV 254, were used for analytical thin-layer chromatography (Merck, Darmstadt, Germany) with visualization under UV (254 and 366 nm), vanillin, and dragendorff. The solvents used were methanol (MetOH), ethyl acetate (AcOEt), dichloromethane (CH_2_Cl_2_), and n-butanol, purchased from Synth (São Paulo, Brazil). Hemodynamic parameters were measured with the Bioamp equipment (Adinstrumentes, Australia) and the Graph Lab software (version 7.0; AD Instruments). An automated rotarod instrument (EFF 411, Insight^®^) and a hot plate (EFF-361, Insight^®^) were used.

### 3.2. Plant Material

The *A. desmanthum* bark was collected in Linhares, Espírito Santo State, Brazil. A voucher specimen was identified and deposited in the herbarium of the Vale Natural Reserve under code CVRD-9470.

### 3.3. Extraction and Isolation

*A. desmanthum* bark (2.5 kg) was extracted with methanol, three times, at room temperature, providing 530 g of crude extract. It was then suspended in H_2_O:MeOH (3:1) and partitioned using dichloromethane, ethyl acetate, and n-butanol. The dichloromethane fraction (10.1284 g) was subjected to repeated silica gel column chromatography (CC) using CH_2_Cl_2_:MeOH, leading to the identification of Aspidocarpine (263.9 mg).

### 3.4. Animals

The tests were carried out with male and female Wistar rats (*Rattus norvegicus*) weighing between 250 and 300 g and male Swiss mice (*Mus musculus*) weighing between 25 and 30 g from the Animal Experimentation Unit of the Universidade Estadual do Norte Fluminense (UEA–UENF), that were kept in an environment with a controlled temperature of 19 °C and a humidity of 50 to 60% for 12 h and with light/dark cycle. Water and food were provided ad libitum. The present study was approved by the Ethics Committee for the Use of Animals of UENF, registered as an additive under protocol number 353, approved on 2 June 2022.

### 3.5. In Vivo Blood Pressure Assessment

Wistar rats were anesthetized by intraperitoneal administration of ketamine (50 mg/k) and xylazine (5 mg/kg) and restrained for insertion of a catheter into the left carotid artery, to measure the following parameters: systolic, median, and diastolic blood pressure. The cannula was heparinized with sodium heparin and 0.9% sodium chloride solution to prevent blood clotting. Another catheter was inserted into the jugular vein for intravenous infusion of the alkaloid Aspidocarpine at a dose of 1 mg/kg (*n* = 4) and 3 mg/kg (*n* = 4), diluted in DMSO, with a volume of 0.1 mL per animal. Prior to the tests, DMSO alone was infused at the same dose to serve as a control in order to eliminate the hypotensive effects of DMSO on the results.

### 3.6. Rotarod Test

The swiss mice (*Mus muscullus*) were previously tested on the rotating bar. Those that fell two or more times in the three-minute period were discarded. After selection of the animals, the alkaloid Aspidocarpine (1 and 3 mg/kg) and the positive control of diazepam (5 mg/kg) were administered intraperitoneally, with a volume of 0.03 mL. Six animals were used per group.

Each individual was placed with all four legs on a rotating bar 8 cm in diameter, 20 cm from the bottom of the equipment, already in motion (20 rpm). The time the mice were able to balance before falling was measured. The mice were observed at the times of 5, 15, 30, and 60 min after sample administration, and remained on the rotating bar for three minutes. At the moment of falling, the chronometer was used to verify the time of equilibrium stopped automatically, the animals were returned to their respective bars, and the chronometer was reactivated, so that the total falls after three minutes could be counted, while a general timer measured the total time of the test (120 min).

### 3.7. Statistical Analysis

All experiments were performed in triplicate, and the results are expressed as mean ± standard error of the mean (SEM). The results obtained were tabulated by the LabChart 7 program, and statistically analyzed through GraphPad Prisma 5. The analysis of variance (ANOVA) was defined, followed by the Newman–Keuls and Bonferroni mean tests, with a reliability index of 95%. Significant difference was taken as * *p* < 0.05, ** *p* < 0.01, and *** *p* < 0.001.

## 4. Conclusions

This work presents the first report on the antihypertensive potential of the alkaloid Aspidocarpine, isolated from the bark of *A. desmanthum*, analyzed by the induction of a decrease in blood pressure and evaluated in vivo. The alkaloid Aspidocarpine showed effective hypotensive activity without causing significant changes in motor coordination. Therefore, this alkaloid may be a promising alternative for the treatment of pathological conditions related to hypertension.

## Figures and Tables

**Figure 1 molecules-27-06895-f001:**
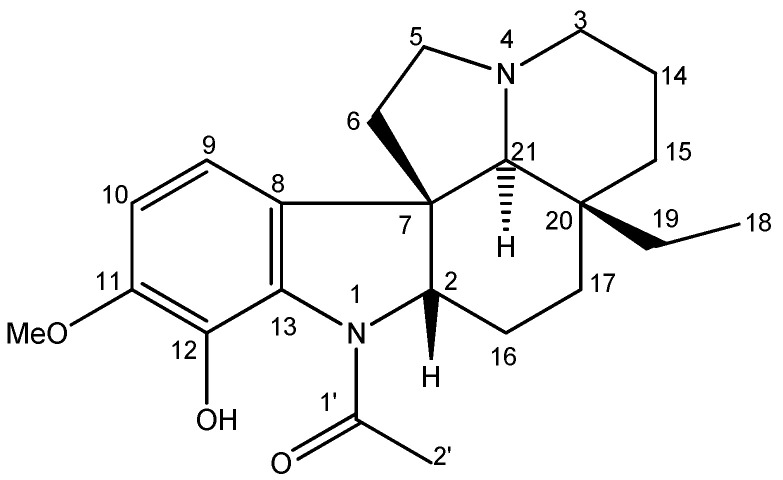
Chemical structure of Aspidocarpine.

**Figure 2 molecules-27-06895-f002:**
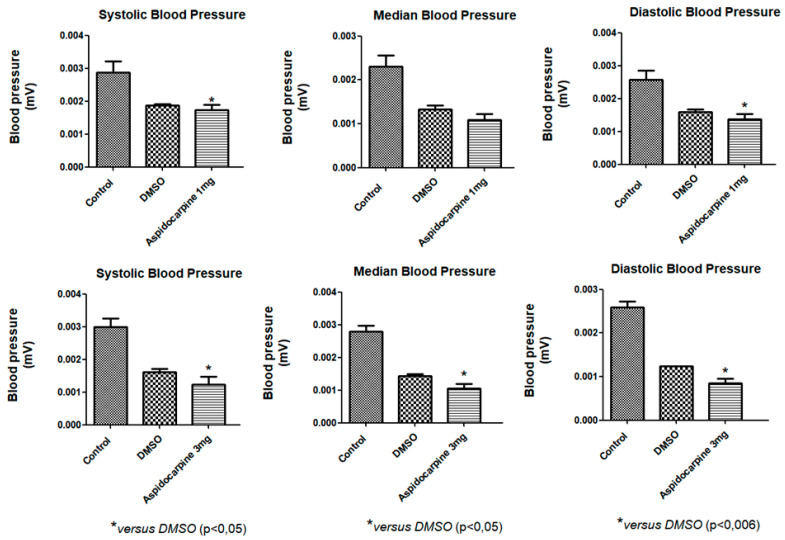
Effects of Aspidocarpine (1 and 3 mg/kg) on blood pressure in Wistar rats. The values are expressed as mean ± SEM (*n* = 4). One-way ANOVA followed by Newman–Keuls test (* *p* < 0.05 compared to control).

**Table 1 molecules-27-06895-t001:** Effects of Aspidocarpine on Swiss mice (*Mus musculus*) in the rotarod test.

	5 min	15 min	30 min	60 min
Control	180.00 s ± 0.00	180.00 s ± 0.00	180.00 s ± 0.00	180.00 s ± 0.00
DZP (5 mg/kg)	178.14 s ± 1.29 *	179.45 s ± 0.94 *	179.47 s ± 0.83 *	179.51 s ± 0.81 *
ASP (1 mg/kg)	179.82 s ± 0.36	180.00 s ± 0.00	179.78 s ± 0.42	179.70 s ± 0.59
ASP (3 mg/kg)	179.62 s ± 0.59	179.26 s ± 0.83	179.95 s ± 0.13	180.00 s ± 0.00

Values are expressed as means ± SEM (*n* = 6). One-way ANOVA followed by Newman–Keuls test (* *p* < 0.05 compared to control). DZP = Diazepam, ASP = Aspidocarpine. Permanence in seconds (s).

## Data Availability

Not applicable.

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
