# Peer review of "Antihypertensive Activity of the Alkaloid Aspidocarpine in Normotensive Wistar Rats"

_molecules, 2022, doi:10.3390/molecules27206895_

Round 1

Reviewer 1 Report

In the present manuscript, the authors characterize chemical and some biologic properties of the alkaloid aspidocarpine. The article is well written, the methods are well described, study results are solid and well presented, and the drawn conclusions are support by the results.

Given the importance of the discovery and characterization of plant derived bioactive compounds, the introduction would benefit from elaborating more on the medical need for research on novel antihypertensive compounds especially with regard to existing drugs.

I suggest to include some ideas on future perspectives in the conclusion section e.g. follow up biologic validation studies but also on how the obtained knowledge could be translated into a potential application in humans.

specific comments:

the negative controls (i.e., control 1 and control 2, Fig.2) should be specified either in the text or the figure legend

Fig.2: median blood pressure: p value is indicated but the stars are not displayed in the graph

Fig.4. please indicate the number of replicates represented in this figure

Fig.S1 and S2: please increase the size/resolution of the figures as it´s impossible to read and interpret the spectra

Reviewer 2 Report

This work presents the first report on the antihypertensive potential of the alkaloid Aspidocarpine, without causing significant changes in motor coordination. Therefore, this alkaloid may be a promising alternative for the treatment of cardiovascular diseases.

Regarding the manuscript I have some comments:

1) Line 52: What is fraction 6?

2) Lines 78-81: the content is more properly for introduction.

3) In Vivo Blood Pressure Assessment

It would be interesting if they made a clearer quantitative description of the results of antihypertensive activity. Controls 1 and 2 are not clearly defined. They do not compare the hypotensive effect of Aspidocarpine respect to previously published extracts.

4) They did not mention how many rats and mice were used in the studies.

5) In the SI 1H and 13C NMR spectra should be bigger

Reviewer 3 Report

Dear authors,

The paper that was given to me to review is very interesting. Natural substances exert many biological activities and represent an inexhaustible source of therapeutic possibilities. On the other hand, hypertension is a socially significant disease whose treatment represents a therapeutic challenge.

I consider the proposed article suitable for publication after major improvements.

To make your article even better, I have a few recommendations. I have also pointed out some inaccuracies and omissions, as follows.

·         I believe that the article is not focused so much on the botanical feature, so it is better to exclude the following keywords – Apocynaceae; alkaloid

·         In my opinion, the use of the words experimental animals and rodents are more appropriate than “individuals”

·         Please reconsider the way you have written of the plant family on line 32

·         It is a good idea to list some of the biological properties you mentioned on line 42

·         The paragraph on lines 43-46 needs rewriting because of the repetitions. The same goes for lines 78-81

·         The information presented in lines 51-55 is appropriate to include in the Methods section, not in Results and Discussion.

·         On line 79, two sources are mentioned but not referenced

·         I would like the authors to specify how the effective doses of the investigated substance were selected. Have pilot toxicity tests been done, or is there any literature on this? In my opinion, since this is the first study of the compound, this information should be included as a basis for the preclinical trials conducted.

·         The title of Fig. 2 says “from four experiments”, the explanation of which is missing in the Method section; Something else about Fig. 2 – it is not clear what are groups Control 1 and Control 2; I recommend that the authors label the six graphs in the figure somehow and include it in the figure title; on the graphs of the first and second row on the left, the inscription "Pressureal" should be redone

·         I suggest that the authors include lines 111-116 in the Introduction section

·         Missing description of results from Fig. 2

·         The rotarod test was done with mice, contrary to the proposed title of the article. Why were mice used when the rotarod test can be used in experimental rats? This also contradicts the title of the article.

·         Fig. 3 - I suggest that the authors present that part of the Y axis that shows the differences in values; the graphics so suggested don't do it well. The title states "six experiments", which is also not described in the methods section

·         I direct the authors to specify the meaning of * symbol - in Fig. 2 it is defined as p<0.05, and in Fig. 3 as p<0.0001. This is not described in the Methods section.

·         The information presented in lines 156-160 is not related to the purpose of the study, namely - the antihypertensive potential of Аspidocarpine

·         About the Hot Plate test - If any, the authors did not clarify the relationship between the presented results on the analgesic properties of the substance and its antihypertensive potential - the aim of the present study. I recommend that this part of the text be excluded from the article. Anyway, Fig. 4 should be revised to show the effects of the substance at the two administered doses compared with that of the referent Diazepam.

·         Materials and Methods – information is missing on how many experimental animals were used, as well as their distribution by group and number in each group; the protocol number on line 225 does not show the date.

·         Why were male and female Wistar rats used in the blood pressure study and only male mice in the Rotarod test? And why mice in the first place? Why is this not commented on in the results? Are there any differences in the effect between male and female animals? This should be commented in the Discussion, or female rats should be excluded from the presented data.

·         I recommend to the authors that the Methods section be rewritten to make it more understandable, specific, and reproducible.

·         I believe that the inclusion of biochemical indicators in this study would contribute to its completeness, and help to hypothesize the possible mechanisms of action of Adipocarpine.

·         Finally, in the Conclusion section: Not all cardiovascular diseases are characterized by elevated blood pressure. A more delicate approach is recommended in conclusions for pilot studies like current, and these should be based on the results obtained. In my opinion "cardiovascular diseases" (lines 277-278) should be changed to "pathological conditions related to hypertension".

Review date

23 Sept 2022

Round 2

Reviewer 3 Report

The Authors agreed with most of the suggested corrections and gave good reasons for their position. I recommend that the article be published in this version.